# A Novel Pathogenic Sense Variant in Exon 7 of the HK1 Gene in a Patient with Hexokinase Deficiency and Gilbert Syndrome

**DOI:** 10.3390/genes15121576

**Published:** 2024-12-07

**Authors:** Magdalena Bartnik, Weronika Pawlik, Beata Burzyńska, Konrad Wasilewski, Elżbieta Kamieńska, Tomasz Urasiński

**Affiliations:** 1Department of Paediatrics Hemato-Oncology and Paediatric Gastroenterology, Pomeranian Medical University, 70-204 Szczecin, Poland; magdalena.bartnik@pum.edu.pl (M.B.); elzbieta.kamienska@pum.edu.pl (E.K.); tomasz.urasinski@pum.edu.pl (T.U.); 2Institute of Biochemistry and Biophysics, Polish Academy of Sciences, 01-447 Warsaw, Poland; atka@ibb.waw.pl (B.B.); konradwasilewski15@gmail.com (K.W.)

**Keywords:** hemolytic anemia, hexokinase deficiency, HK1 gene, next-generation sequencing, children

## Abstract

Background: Hexokinase (HK) deficiency is a rare autosomal recessively inherited disease manifested by chronic nonspherocytic hemolytic anemia. Most patients present with a mild to severe course of the disease (fetal hydrocephalus, neonatal hyperbilirubinemia, severe anemia). We reviewed 37 cases of patients with hexokinase deficiency described so far, focusing on the severity of the disease, clinical presentation, treatment applied, and genetic test results. Methods: We present a 10-year-old girl who initially presented with symptoms of weakness, excessive fatigue, and yellowing of the skin and sclerae. Genetic testing detected the (TA)7 variant in both alleles of the UGT1A1 gene and diagnosed Gilbert’s disease. In the follow-up, red cell hemolysis was observed. The diagnosis was extended, and tests for red cell enzymopathy were performed and a reduced level of hexokinase—0.65 IU/gHb (normal 0.78–1.57) was found. Next-generation sequencing revealed a new sense-change variant in exon 7 in the hexokinase gene not previously reported in databases. Results: Up to this date, only around 37 cases of hexokinase deficiency associated with hereditary nonspherocytic hemolytic anemia have been documented around the world. Diagnosing hexokinase deficiency involves clinical evaluation, laboratory testing, and genetic analysis. Management focuses on treating symptoms and preventing complications; there is no cure for the underlying enzyme deficiency. In patients with severe anemia, the treatment is multiple blood transfusions followed by iron chelation therapy. Conclusions: Understanding and diagnosing hexokinase deficiency is critical for providing appropriate care and improving the quality of life for affected individuals.

## 1. Introduction

Hexokinase (HK) deficiency is a rare autosomal recessive inherited metabolic disorder that manifests itself as nonspherocytic hemolytic anemia. Hexokinase is a key enzyme in the first step of glycolysis—the phosphorylation of glucose to glucose-6-phosphate (G6P), a metabolic pathway capable of delivering adenosine triphosphate (ATP) [1]. The ATP generated in glycolysis is used as an immediate energy source for various cellular processes (e.g., for red blood cells). Hexokinase is present in almost all tissues, but its activity varies. In tissues like muscles and the brain, where energy demand is high, hexokinase ensures a continuous supply of G6P for energy production [2]. Red blood cells lack mitochondria and depend entirely on glycolysis for ATP production. In hexokinase deficiency, the reduced conversion of glucose to G6P leads to decreasing ATP production and a failure to maintain red blood cell integrity, causing nonspherocytic hemolytic anemia It and low levels of 2,3-bisphosphoglycerate (2,3-BPG), reducing oxygen delivery to tissues, since 2,3-BPG decreases hemoglobin’s oxygen affinity [3]. The enzyme hexokinase is encoded by the HK1 gene located on chromosome 10q22.1. The HK1 gene expresses different isoforms of hexokinase, with the RBC-specific isoform being crucial for red blood cell metabolism [4].

When hexokinase is deficient or non-functional, cells, especially red blood cells, cannot metabolize glucose properly, and this leads to a variety of symptoms. These symptoms often include chronic hemolytic anemia, as red blood cells are particularly dependent on glycolysis for energy. Most patients present with a mild onset of the disease and may experience fatigue, jaundice, and an increased risk of infections. Some individuals may present a severe course of the disease (fetal hydrocephalus, neonatal hyperbilirubinemia, severe anemia), which requires chronic blood transfusions. Severe manifestation is more likely when hexokinase activity is critically reduced, when triggers such as infections or oxidative stress exacerbate hemolysis, or splenic hyperactivity worsens anemia and hemolysis. Early recognition, supportive care, and close monitoring are essential in managing severe cases of hexokinase deficiency.

Hexokinase I (HK-I) is the main form of the enzyme found in tissues such as the brain, muscles, and red blood cells that rely heavily on glucose for their normal function. Red blood cells contain a distinct variant of Hexokinase I, known as HKr, which is encoded by the HK-I gene [4]. Up to this date, approximately 37 cases of this disease have been reported worldwide, and only a few of them have been diagnosed on a molecular level [5]. We now report a novel pathogenic sense variant in exon 7 of the HK1 gene in a patient with coexistent Gilbert syndrome. This is the third time a case of the occurrence of these two disease entities in one patient has been described.

## 2. Case Presentation

### 2.1. Patient Clinical Evaluation

The index case is a 10-year-old female patient who was admitted to the Pediatric Gastrology Department with a suspicion of Gilbert syndrome. On admission, the girl was in good general condition. She was awake and alert, and her circulatory and respiratory functions were efficient. The only abnormality on physical examination was visible yellowing of the sclerae. During an interview, she reported yellowing of the sclera and skin from the neonatal period and a single episode of vomiting a few months before admission, after which yellowing of the sclera was observed by a general practitioner. Laboratory tests performed on an outpatient basis before admission revealed hyperbilirubinemia. Total bilirubin was 3.56 mg/dL (bound bilirubin level was 1.03 mg/dL). The liver and pancreatic function parameters were normal, and hepatitis B and C results were negative. Moreover, abdominal ultrasonography revealed small deposits in the gallbladder. In the family history, the mother had cholecystectomy at 28 years of age because of cholelithiasis, the grandmother had cholecystectomy at about 30 years of age and kidney stones, and the father had symptoms that might indicate Gilbert’s syndrome.

During hospitalization, laboratory test showed hyperbilirubinemia—2.77 mg/dL (the bound bilirubin level was 0.88 mg/dL).

Genetic screening, using the Sanger DNA sequencing technique, detected the (TA)7 variant in both alleles of the UGT1A1 gene. A Gilbert syndrome diagnosis was confirmed. Treatment with ursodeoxycholic acid was ordered due to asymptomatic cholelithiasis. The patient was discharged in good general condition.

After one and a half years, she was readmitted to the Pediatric Hemato-Oncology Department on a scheduled basis to expand the hematological diagnosis. On admission, her skin and sclerae were slightly yellowed. In blood analysis, despite normal red cell parameters, hyperbilirubinemia with free bilirubin predominance persisted, and an elevated reticulocyte percentage (from 3.4 to 4.23%) and decreased haptoglobin were observed, confirming red cell hemolysis. The Coombs test was negative. The diagnosis was extended with the EMA test, which was negative. The results of morphology and biochemistry of our patient are listed in the tables below (Table 1 and Table 2).

Tests for red cell enzymopathy showed hexokinase levels below reference values—0.65 IU/gHb (normal range—0.78–1.57) and pyruvate kinase levels above reference values —21.94 IU/gHb (N = 6.87–20.53). As there was no anemia noted to date, the study of hexokinase was repeated with a comparable result—0.60 IU/gHb. Other tests checking the enzyme activity for glucose-6-phosphate dehydrogenase (G6PD), isomerase, phosphofructokinase (PFK), aldolase, phosphoglycerate kinase (PGK 1), and methemoglobin reductase were normal (Table 3).

### 2.2. Molecular Analysis of the HK1 Gene

The diagnosis was enhanced by sequencing of the HK1 gene. gDNA was isolated from whole blood samples and used to amplify the promoter coding and flanking regions of the HK1 gene by PCR. Sequences of primers and reaction condition are available upon request. The obtained PCR products were purified using the QIAquick PCR Purification Kit (Qiagen, Hilden, Germany) and sequenced directly with BigDye Terminators and appropriate primers using an ABI Prism 377 sequencer (Applied Biosystems, Foster City, CA, USA). Analysis of the genomic sequences of exons 1–18 of the HK 1 gene in the proband revealed a new missence variant in exon 7 (chr10: 71129298G>A, NM_033496, NP_277031.1, c.790G>A, cDNA.891G>A, g.99559G>A, p.Gly265Arg) in the hexokinase gene. The potential pathogenicity of the detected variant was estimated using MutationTaster, PROVEAN and Mendelian Clinically Applicable Pathogenicity (M-CAP) software [6,7,8]. The identified variant was cross-referenced with a population database (gnomAD, https://gnomad.broadinstitute.org/ accessed on 15 August 2024), a single nucleotide polymorphism database (dbSNP, https://www.ncbi.nlm.nih.gov/snp/), and ClinVar (https://www.ncbi.nlm.nih.gov/clinvar/ accessed on 15 August 2024); a bioinformatic analysis predicted deleterious effects where this mutation was detected. The c.790G>A mutation is not reported in databases (Table 4).

A final diagnosis of nonspherocytic hemolytic anemia due to hexokinase deficiency was made, and the patient was referred to the hematology outpatient clinic for further observation.

## 3. Discussion

Hexokinase deficiency in human red blood cells was first linked to hereditary nonspherocytic hemolytic anemia by Valentine et al. in 1967 [9]. It is an extremely rare red blood cell enzyme disorder, and since Valentine et al.’s report, only around 37 other variants of hexokinase deficiency associated with hereditary nonspherocytic hemolytic anemia have been documented around the world. Up to this date, no cases have been reported from Poland at either the phenotypic or the genetic level. In this paper, we present a Polish patient experiencing mild chronic hemolysis with hexokinase deficiency, in whom we examined the genetic cause of the disease.

Furthermore, we reviewed to the best of our knowledge all the cases of patients with hexokinase deficiency described so far, focusing on the severity of the disease, its clinical presentation, the treatment applied, and genetic test results. The results of the analysis are presented in Table 5.

Diagnosing hexokinase deficiency involves several steps, combining clinical evaluation, laboratory testing, and genetic analysis. Clinical evaluation involves an assessment of the patient’s medical history, symptoms, and any family history of hemolytic anemia or related metabolic disorders. Frequently occurring manifestations of the disease are collected and shown in Table 4. The most frequently observed symptom was hemolytic anemia. Of the thirty-seven patients reviewed, the fifteen of them (41%), had a severe form of hemolytic anemia, often requiring continuous blood transfusions. Nine of them (24%) presented a mild pattern of chronic hemolytic anemia, and only six patients (16%) had either a pattern of compensated HA or an asymptomatic disease course. In seven patients, the severity of the anemia was unknown due to a lack of availability of the literature. What is more, jaundice was observed in twelve patients (32%) and spleen enlargement in eleven of them (30%), and psychomotor developmental delay was seen in nine patients (24%). Other less common symptoms presented by patients with hexokinase deficiency included fatigue and cholelithiasis. In analyzing patients’ laboratory tests, we could frequently observe hemolysis markers such as anemia, elevated reticulocytes, elevated bilirubin levels, LDH, and low haptoglobin. All the above-mentioned symptoms and abnormalities in laboratory tests should prompt clinicians to widen the diagnosis to include testing for red cell enzymopathy or genetic testing for HK1 gene mutations.

The differential diagnosis for hexokinase deficiency includes other conditions that cause hemolytic anemia. Key differentials include enzyme deficiencies in glycolysis or red blood cell metabolism (e.g., pyruvate kinase deficiency, glucose-6-phosphate dehydrogenase (G6PD) deficiency, and phosphoglucose and triosephosphate isomerase deficiency), hemoglobinopathies, membrane disorders, autoimmune hemolytic anemia, and drug-induced hemolysis.

It is worth noting that in the case of our patient, anemia was not observed throughout the entire follow-up period. The main symptom of concern due to which the diagnosis was widened was yellowing of the skin. In further research, a Gilbert syndrome (GS) diagnosis was confirmed with genetic screening. To date, only two other cases of a coincidence of Gilbert syndrome and hexokinase deficiency have been described by Koralkova P et al. in 2016 [28]. These patients presented with jaundice and mild hemolysis with normal hemoglobin levels, and the diagnosis was expanded because of a family history of hexokinase deficiency. In our case, the diagnosis was expanded due to persistent hemolytic features despite the recognition of GS.

To search for congenital causes of hemolytic anemia, further diagnostic processes include Coombs and EMA tests and red cell enzymatic assays. In the case of our patient, a double enzyme assay showed a decrease in hexokinase with normal values of other enzymes, which identified a hexokinase deficiency. However, due to the low availability of enzyme tests, it is likely that most patients with mild forms of the disease are not properly diagnosed. Furthermore, recent reports presented by Ukonmaanaho et al. show that there are cases of patients with severe hemolytic anemia requiring regular blood transfusions whose enzymatic activity test results did not show any abnormalities (their hexokinase level was in the normal range), while changes in the kinetic properties of hexokinase were observed, and genetic testing showed alterations in the HK1 gene that may be responsible for the clinical presentation of the disease [33]. This points to the necessity of extending diagnostics to include tests measuring hexokinase activity, the conducting of kinetic analyses, and, if available, genetic testing in patients with unknown cause of hemolytic anemia. Combining these diagnostic methods provides a comprehensive assessment of hexokinase deficiency, ensuring accurate diagnosis and appropriate management of the condition.

Up to the present date, nineteen cases of hexokinase deficiency have been characterized by demonstrating mutations in the genome. Most of them are listed in Table 5. In our patient, sequencing of the HK1 gene was performed. A sense-change variant in exon 7 (chr10: 71129298G>A, NM_033496, NP_277031.1, c.790G>A, cDNA.891G>A, g.99559G>A, p.Gly265Arg) was detected. This variant has not been previously reported in databases (gnomAD, ClinVar). Most sense-change variants reported in the ClinVar database are classified as potentially pathogenic variants. Molecular testing for hexokinase deficiency is possible but not always easy to perform due to the complexity of the HK1 gene, the rarity of the condition, and the need for functional validation. Enzyme activity assays typically precede molecular testing and guide its necessity. Advances in genetic technology, like NGS, are improving accessibility and efficiency, but testing is still not routine in many settings.

The management of hexokinase deficiency focuses on treating symptoms and preventing complications; currently, there is no cure for the underlying enzyme deficiency. In patients with severe hemolytic anemia, the principal treatment was multiple blood transfusions during periods of exacerbated hemolysis followed by iron chelation therapy. In addition, many of these patients (22%) underwent splenectomy with good results, reducing or eliminating the need for further transfusions. On the other hand, some patients continue to require continuous transfusions of red cell concentrate even after spleen removal surgery. Khazal et al. reported the first case of successful allogeneic bone marrow transplant in a patient with hemolytic anemia secondary to hexokinase deficiency from an unaffected histocompatible sibling donor [27]. One year after bone marrow transplantation, the patient remains transfusion-independent; however, his dropping donor chimerism remains concerning for graft rejection.

Due to the extreme rarity of this disease worldwide, each case described gives additional information about the possible course and expected complications. For clinicians, such information is essential to make treatment decisions that are best for the patient’s health and life. This case is one of the examples showing a mild presentation of the disease and demonstrating a model diagnostic process including a test for red blood cell enzymopathy and genetic testing.

## 4. Conclusions

The discovery of a novel sense-change mutation in the HK1 gene furthers the understanding of the enzymatic properties of hexokinase. This discovery highlights the significant role that the study of hexokinase-deficient individuals plays in better understanding the key function of hexokinase in the glycolytic pathway, particularly in red blood cell metabolism. The molecular analysis of these defects has now become routine, enabling proper diagnosis of hemolytic anemia and offering the potential for a cure through gene therapy. There is still need for further research in terms of the correlation between genotype and phenotype in this condition. A study on a larger cohort is needed to draw statistically significant conclusions. Understanding and diagnosing hexokinase deficiency is critical for providing appropriate care and improving the quality of life for affected individuals.

## Figures and Tables

**Table 1 genes-15-01576-t001:** Results of our patient—morphology.

	Normal Range	Result No. 1	Result No. 2	Result No. 3	Result No. 4	Result No. 5	Result No. 6	Result No. 7	Result No. 8
RBC (mL/uL)	4.10–5.10	4.65	4.53	4.17	4.06	4.11	3.95	4.09	4.28
Hemoglobin (g/dL)	12.0–15.5	13.5	13.6	12.5	12.5	12.7	12.4	12.9	13.4
Hematocrit (%)	35.0–44.0	40.3	41.0	35.7	36.5	37.4	35.9	38.0	39.5
MCV (fL)	77.0–94.0	86.7	90.5	85.6	89.9	91.0	90.9	92.9	92.3

**Table 2 genes-15-01576-t002:** Results of our patient—biochemistry.

	Normal Range	Result No. 1	Result No. 2	Result No. 3	Result No. 4	Result No. 5	Result No. 6
Reticulocyte count (%)	0.76–2.21	3.9	3.44	4.23	3.75	4.34	
Haptoglobin (g/L)	0.30–2.00	< 0.1	0.15				
Lactate dehydrogenase (U/L)	0–308	144	135	148			
Total bilirubin (mg/dL)	0.00–1.10	3.56	2.77	5.15	5.10	4.59	3.76
Conjugated bilirubin (mg/dL)	0.00–0.30	1.03	0.88	0.52	0.48	0.45	0.69

**Table 3 genes-15-01576-t003:** Enzyme activities of our patient.

Enzyme	Result (IU/gHb)	Normal Value (IU/gHb)
glucose-6-phosphate dehydrogenase	7.47	5.9–10.5
pyruvate kinase	21.94	6.87–20.53
isomerase	27.28	22.40–45.22
phosphofructokinase	3.59	3.36–10.15
aldolase	1.72	1.65–3.44
phosphoglycerate kinase	311.14	121.58–379.46
methemoglobin reductase	23.36	17.54–31.82

**Table 4 genes-15-01576-t004:** Next-generation sequencing results of our patient.

Exon	Details
1	No changes
2	Homozygote rs.1133189 benign polymorphism
3	No changes
4	No changes
5	No changes
6	No changes
7	Missense change (chr10: 71129298G>A, NM_033496, NP_277031.1 c.790G>A, p.Gly265Arg)
8	No changes
9	No changes
10	Homozygote rs.748235 benign
11	No changes
12	No changes
13	TT insertion before exon rs.3086686 benign
14	No changes
15	Change in intron after exon-homozygote rs.2278745 benign
16	No changes
17	Homozygote rs.1227938 benign
18	No changes

**Table 5 genes-15-01576-t005:** Summary of clinical, laboratory, and genetic findings of patients with hexokinase deficiency.

Reference	Age (Years)	Sex	Symptoms	Hb (g/dL)	Reticulocytes (%)	Total Bilirubin	Red Cell HK Activity	Other Procedures	Genetic Testing
Valentine WN et al., 1967 [9]	0 (5 months)	F	pallor/jaundice, hepatomegaly, splenomegaly	9.4	13	N/A	63%	blood transfusions, splenectomy	N/A
Keitt AS et al., 1969 [10]	38	F	jaundice, mild anemia, splenomegaly	N/A	6.1–13	N/A	79%	splenectomy	N/A
K Moser et al., 1970 [11]	22	M	No data available	N/A	1.7–6.3	N/A	49%	N/A	N/A
Necheles TF et al., 1970 [12]	2	M	severe HA	6.5	5	N/A	75%	splenectomy	N/A
Goebel KM et al., 1972 [13]	28	F	paleness, no hepatosplenomega-ly	8.5–9.6	2	N/A	0.59 U/gHb		N/A
Semenuk M et al., 1975 [14]	N/A								
Rijksen G et al., 1978 [15]	30	F	chronic hemolysis	11.3	39	N/A	0.7 U/gHb	blood transfusions, splenectomy	N/A
Board PG et al., 1978 [16]	2	M	anemia, cholelithiasis, neonatal jaundice, hepatomegaly, splenomegaly	8.6–9.4	6.7–8.5	6.1	0.65 U/gHb	N/A	N/A
Beutler E et al., 1978 [17]	11	F	fatigue, mild chronic HA	11.6–12.2	5.1–7.4	1	0.83 U/gHb	N/A	N/A
Gilsanz F et al., 1978 [18]	9	F	anemia, jaundice, mild mental retardation	8.7–9.5	10.2–16.4	5.3	68%	N/A	N/A
Siimes MA et al., 1979 [19]	1	F	anemia	7–9.2	3.1–8.1	N/A	48%	N/A	N/A
Newman P et al., 1970 [20]	19	M	fatigue, jaundice, splenomegaly, abdominal discomfort,	13.8	33	37	0.62 U/gHb	N/A	N/A
Paglia DE et al., 1981 [21]	7	M	anemia, jaundice, hepatomegaly, splenomegaly	9.7	3.6–8.4	8.1	70%	blood transfusion	
Rijksen G et al., 1983 [22]	19	F	neonatal jaundice, severe HA	9.8	50	N/A	25%	splenectomy	homozygous c.2039C>G (Thr680Ser) missense mutation in HK1
Magnani M et al., 1985 [23]	27	F	anemia, hepatomegaly, splenomegaly, sclera jaundice, fatigue	5.3	1	0.5	0.87 U/gHb	blood transfusions, splenectomy	
Magnani M et al., 1985 [24]	1	M	generalized convulsions, hepatomegaly, splenomegaly, psychomotor retardation	11.6	3	0.3	45%	blood transfusions,	
Kanno H et al., 1997 [25]	29 weeks (GA)	F	IUGR fetus with severe HA, periventricular leucomalacia	3.7	42	N/A	17%	decease	
de Vooght KM et al., 2009 [26]	33	M	jaundice, fatigue, inability to concentrate, splenomegaly	8.8	79	73	0.79 U/gHb	N/A	−193A>G mutation in the erythroid-specific promoter of *HK1* and exon 3 c.278G>A missense mutation
Sajad Khazal et al., 2016 [27]	4	M	hydrops fetalis, severe anemia	4.5–9	N/A	N/A	0.7 U/gHb	regular blood transfusions, splenectomy, bone marrow transplant	homozygous nucleotide substitution in the first nucleotide of exon 13 of the HK 1 gene
Koralkova P et al., 2016 [28]	2	M	severe hemolysis	7.9	8.8	54	0.64 U/gHb	blood transfusions	heterozygous for mutations c.−193A N G and c.873-2A N G
	4	M	compensated HA with no clinical symptoms	9.5	6.2	62	0.58 U/gHb	N/A	heterozygous for mutations c.−193A N G and c.873-2A N G
	8 months	M	severe HA, psychomotor retardation, secondary epilepsy (result of hypoxia and bleeding during delivery)	7.4	6.3	120	0.71 U/gHb	blood transfusions	homozygous for the p.(Arg93Gln) mutation
	2	M	severe HA	8.6	4.7	124	1.14 U/gHb	blood transfusions	c.2599C N T p.(His867Tyr)
	22	M	jaundice	16	6.0	24	0.65 U/gHb	Gilbert’s syndrome	heterozygous for the p.(Thr600Met) mutation
	12	F	mild hemolysis	13.5	3.1	100	0.37 U/gHb	Gilbert’s syndrome	heterozygous for the c.493-1G N A mutation
	12	F	asymptomatic, mild methemoglobinem-ia	13.7	1.9	17.6	0.33 U/gHb	G6P deficiency	heterozygous c.278A N G p.(Arg93Gln), homozygous for a mutation in G6PD: c.477G N C p.(Met159Ile)
Sonaye Ruhi et al., 2018 [29]	21	F	vomiting, loose motion, yellowish sclera, hepatomegaly, splenomegaly	6.7	N/A	12.58	N/A	blood transfusion	N/A
Manu Jamwal et al., 2019 [30]	9 months	M	severe anemia, jaundice, delayed milestones	4.7	N/A	N/A	N/A	several blood transfusions starting with an exchange transfusion at day 4 of life, death at age 1	homozygous nonsense variant NM_033496.2:c34C>T, p.Arg12Ter, in exon 1 of HK1
Rashmi Dongerdiye et al., 2021 [31]	4	F	severe anemia, developmental delay	3.5	N/A	N/A	N/A	regular blood transfusions	homozygous variant in HK gene c.2714C>A (p.Thr905Lys)
Sasaki E et al., 2022 [32]	two siblings and one cousin (dizygotic twin I)		deceased, additional malformations coexistent 16p13.11 microdeletion syndrome	N/A	N/A	N/A	N/A	N/A	homozygous c.278G>A p(Arg93Gln) in exon 3 of HK 1
			HA, developmental delay, epilepsy, microcephaly, white matter atrophy, recurrent apnoeic episodes	N/A	N/A	N/A	N/A	N/A	
				N/A	N/A	N/A	N/A	N/A	
	half-brother		HA, developmental delay, white matter atrophy	N/A	N/A	N/A	N/A	N/A	died without genetic testing
EM Ukonmaanaho et al., 2024 [33]	3 months	F	severe HA	6.7	7.3	33	1.1 U/gHb	blood transfusions, HSCT planned	homozygous for the *HK1* variant c.2599C>T, p.(His867Tyr)
	30	M	Mild chronic HA, splenomegaly	12.1	4.6	103	0.8 U/gHb	splenectomy	heterozygous for the *HK1* variant c.2599C>T, p.(His867Tyr) and c.-193A>G
	37	M	Mild chronic HA, splenomegaly	15.9	0.7	14	0.52 U/gHb	cholecystectomy	heterozygous for the *HK1* variants c.2361_2362del, p.(Gln788Aspfs*4) and c.-193A>G
	26	M	Cholelithiasis, compensated hemolysis	11.9	3.9	175	N/A	N/A	heterozygous for the *HK1* variants c.372+1G>A and c.-193A>G

## Data Availability

The original contributions presented in the study are included in the article, further inquiries can be directed to the corresponding author.

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
