# Peer review of "A Novel Pathogenic Sense Variant in Exon 7 of the HK1 Gene in a Patient with Hexokinase Deficiency and Gilbert Syndrome"

_genes, 2024, doi:10.3390/genes15121576_

Round 1

Reviewer 1 Report

Comments and Suggestions for Authors

The authors present a novel pathogenic sense variant in exon 7 of the HK1 gene in a patient with hexokinase deficiency and Gilbert syndrome. The manuscript is interesting and i have the following concerns:

1. What are the clinical data that lead the physicians screen for the HK1 gene? Please add a paragraph to the discussion. 

2. What is the differential diagnosis of the hexokinase deficiency? Please provide an educational table with similar diseases that the clinicians should take into account. 

3. In which cases are the clinical manifestations of the hexokinase deficiency more severe?

4. Is the molecular testing for hexokinase deficiency easy to perform?

Comments on the Quality of English Language

There are errors in the whole text and especially in the abstract. Please provide a language polishing certificate for the whole manuscript. 

Author Response

The response to Reviewer No 1:

Thank you for the revision and your favorable judgment of the manuscript. We followed your recommendations, and we added more information’s and details to our manuscript:

  1. Line 174: “All the above-mentioned symptoms and abnormalities in laboratory tests should prompt clinicians to widen the diagnosis to include testing for red cell enzymopathy or genetic testing for HK1 gene mutations.”
  2. We developed the topic of differential diagnosis in the discussion. However, as most of these diseases present with similar symptoms and the diagnosis of these disease entities includes testing for red cell enzymopathy and simple laboratory tests, we believe that too little clinically relevant data would be included in the table.

We added line 177: “The differential diagnosis for hexokinase deficiency includes other conditions that cause hemolytic anemia. Key differentials include enzyme deficiencies in glycolysis or red blood cell metabolism (e.g. pyruvate kinase deficiency, glucose-6-phosphate dehydrogenase (G6PD) deficiency, phosphoglucose and triosephosphate isomerase deficiency), hemoglobinopathies, membrane disorders, autoimmune hemolytic anemia and drug-induced hemolysis.”

  1. We added information in line 55 “Severe manifestation is more likely when HK activity is critically reduced, when triggers such as infections or oxidative stress exacerbate hemolysis, or splenic hyperactivity worsens anemia and hemolysis. Early recognition, supportive care, and close monitoring are essential in managing severe cases of hexokinase deficiency.”

  1. Line 213 “Molecular testing for hexokinase deficiency is possible but not always easy to perform due to the complexity of the HK1 gene, the rarity of the condition, and the need for functional validation. Enzyme activity assays typically precede molecular testing and guide its necessity. Advances in genetic technology, like NGS, are improving accessibility and efficiency, but testing is still not routine in many settings.”

Reviewer 2 Report

Comments and Suggestions for Authors

This is an interesting and generally well-written case report. The extensive review of the 37 cases of the literature is a very positive part. However, some things could be improved:

1)      Please elaborate in the introduction regarding the function of hexokinase in the glycolytic pathway, and the genetics of condition of the hexokinase deficiency.

2)      Line 173: please elaborate on the other two cases and compare with this case.

3)      Please elaborate more in the discussion on the clinical significance of reporting this case.

4)      The case report could be better organized. Sections 2 and 3 could be merged in a single “Case presentation” section and the subtitles could be kept.  

5)      English need careful editing as in many cases there are syntax/grammar errors, eg lines 73-78 (need rewriting to be improved), line 99 (“were” instead of “was”), line 211 (erase one “after”).

6)      Make sure to give only abbreviations that are used in the main text, eg HK is not used in the text, but the term hexokinase instead is always used.

Author Response

Thank you very much indeed for all your remarks and comments. All of them were considered, and respective corrections were introduced into the body of the manuscript.

  1. Elaboration regarding the function of hexokinase and genetics of hexokinase deficiency.

Line 33-48: Hexokinase (HK) deficiency is a rare autosomal recessive inherited metabolic dis-order that manifests itself as nonspherocytic haemolytic anemia. Hexokinase is a key enzyme in the first step of glycolysis - the phosphorylation of glucose to glu-cose-6-phosphate (G6P), a metabolic pathway capable of delivering adenosine triphos-phate (ATP) [1]. The ATP generated in glycolysis is used as an immediate energy source for various cellular processes (e.g. for red blood cells). Hexokinase is present in almost all tissues, but its activity varies. In tissues like muscles and the brain, where energy demand is high, hexokinase ensures a continuous supply of G6P for energy production [2]. Red blood cells lack mitochondria and depend entirely on glycolysis for ATP production. In hexokinase deficiency, the reduced conversion of glucose to G6P leads to decreasing ATP production, failure to maintain red blood cell integrity, causing nonspherocytic hemolytic anemia and low levels of 2,3-bisphosphoglycerate (2,3-BPG), reducing oxygen delivery to tissues since 2,3-BPG decreases hemoglobin's oxygen affinity. [3] The enzyme hexokinase is encoded by the HK1 gene, located on chromosome 10q22.1. The HK1 gene expresses different isoforms of hexokinase, with the RBC-specific isoform being crucial for red blood cell metabolism. [4]

[3] van Wijk, Richard, and Wouter W van Solinge. “The energy-less red blood cell is lost: erythrocyte enzyme abnormalities of glycolysis.” Blood vol. 106,13 (2005): 4034-42. doi:10.1182/blood-2005-04-1622

  1. Due to the lack of clinical data in Koralkov's publication, we decided not to compare these cases with our case. The only linking factor was the diagnosis of Gilbert syndrome, which seems to be a coincidental finding.

  1. Line 231: “Due to the high rarity of this disease worldwide, each case described gives additional information about the possible course and expected complications. For clinicians, such information is essential to make treatment decisions that are best for the patient's health and life. This case is one of the examples showing a mild presentation of the disease and demonstrating a model diagnostic process including a test for red blood cell enzymopathy and genetic testing.”

  1.  

  1. Correction of lines 73-78:

Genetic screening, using the Sanger DNA sequencing technique, detected the (TA)7 variant in both alleles of the UGT1A1 gene. Gilbert syndrome diagnosis was confirmed. Treatment with ursodeoxycholic acid was ordered due to asymptomatic cholelithiasis. The patient was discharged in good general condition.

Correction of lines 98-99:

phosphofructokinase (PFK), aldolase, phosphoglycerate kinase (PGK 1), methemoglobin reductase were normal [Tab. 3].

Correction of lines 210-212:

One year after the transplantation the patient remains transfusion-independent however his dropping donor chimerism remains concerning for graft rejection.

  1. Abbreviation HK was changed to hexokinase in all places.

Reviewer 3 Report

Comments and Suggestions for Authors

The manuscript is valuable to the sparse literature on hexokinase deficiency, particularly its genetic underpinnings. While the study is meticulously documented, deeper molecular insights, functional validation, and a broader discussion of therapeutic implications would significantly enhance its impact. These improvements could serve as a benchmark for future investigations into rare enzymopathies.

Comments: 

1. While the study acknowledges the key role of hexokinase in glycolysis, it lacks depth in explaining the biochemical and structural implications of the identified variant on enzyme function

2. The discussion on treatment remains generic, focusing on symptomatic management. Emerging therapeutic strategies, such as gene therapy or enzyme replacement therapy, could have been explored.

3. The study is a single case report, and while this is understandable given the rarity of the disease, the conclusions about genotype-phenotype correlation are speculative and require validation in larger cohorts.

4. While bioinformatics tools are valuable, the study lacks functional validation of the mutation's effect, such as enzyme kinetics or structural modelling.

5. Some sections reiterate information already presented, such as the rarity of hexokinase deficiency and its associated clinical features, which could have been summarized more succinctly

6- Although extensive, many citations are outdated. The inclusion of recent advances in molecular hematology and glycolytic enzyme disorders would have enriched the manuscript

Author Response

Thank you very much indeed for all your remarks and comments. All of them were considered, and respective corrections were introduced into the body of the manuscript.

  1. We enriched the introduction in more complex explanation of the role of hexokinase in glycolysis, and implications of the identified variant on enzyme function.

Lines 33-48: “Hexokinase (HK) deficiency is a rare autosomal recessive inherited metabolic dis-order that manifests itself as nonspherocytic haemolytic anemia. Hexokinase is a key enzyme in the first step of glycolysis - the phosphorylation of glucose to glu-cose-6-phosphate (G6P), a metabolic pathway capable of delivering adenosine triphos-phate (ATP) [1]. The ATP generated in glycolysis is used as an immediate energy source for various cellular processes (e.g. for red blood cells). Hexokinase is present in almost all tissues, but its activity varies. In tissues like muscles and the brain, where energy demand is high, hexokinase ensures a continuous supply of G6P for energy production [2]. Red blood cells lack mitochondria and depend entirely on glycolysis for ATP production. In hexokinase deficiency, the reduced conversion of glucose to G6P leads to decreasing ATP production, failure to maintain red blood cell integrity, causing nonspherocytic hemolytic anemia and low levels of 2,3-bisphosphoglycerate (2,3-BPG), reducing oxygen delivery to tissues since 2,3-BPG decreases hemoglobin's oxygen affinity. [3] The enzyme hexokinase is encoded by the HK1 gene, located on chromosome 10q22.1. The HK1 gene expresses different isoforms of hexokinase, with the RBC-specific isoform being crucial for red blood cell metabolism. [4]”

[3] van Wijk, Richard, and Wouter W van Solinge. “The energy-less red blood cell is lost: erythrocyte enzyme abnormalities of glycolysis.” Blood vol. 106,13 (2005): 4034-42. doi:10.1182/blood-2005-04-1622

  1. As with many diseases, including inborn errors of metabolism, enzyme therapy and gene therapy appear to be the treatments of the future. One disease in which enzyme therapy plays a key role is Gaucher disease. Research into new treatment options and into hexokinase deficiency appears to be important, however we did not find any relevant publication concerning gene therapy in this disease.

  1. Thank you for this comment, I agree with the reviewer. We added a line 245 in conclusions: A study on a larger cohort is needed to draw statistically significant conclusions.

  1. We did not perform enzyme kinetics nor structural modeling. While it is more common to analyze enzyme kinetics in broader studies involving multiple samples or comparisons, however we know that studying enzyme kinetics in a single case is both feasible and clinically informative. It is possible that in future research we will consider extending the study to include these aspects.

  1. Due to the small number of cases, we believe that a larger summary and comparison of the clinical picture of the disease would not be concise enough. Studies on a larger number of cases are needed to be able to draw concrete conclusions.

  1. Thank you for this comment. However, most modern publications on the subject have been quoted. Due to the rarity of the disease, there are not many papers on the subject.

Round 2

Reviewer 1 Report

Comments and Suggestions for Authors

I have no further concerns.